
**Assessing tropical cyclone compound flood risk using hydrodynamic**
**modelling: a case study in Haikou City, China**
**Qing Liu, Hanqing Xu, Jun Wang***
Key Laboratory of Geographic Information Science of Ministry of Education, School of
Geographic Science, East China Normal University, Shanghai, 200241, China
*Corresponding author.
Email address: uniqliu@163.com (Q. Liu), xuhq@stu.ecnu.edu.cn (H.Q. Xu),
jwang@geo.ecnu.edu.cn (J. Wang).
**Abstract** The co-occurrence of storm tide and rainstorm during tropical cyclones (TCs) can lead
to compound flooding in low-lying coastal regions. The assessment of TC compound flood risk
can provide vital insight for research on coastal flooding prevention. This study investigates TC
compound flooding by constructing a storm surge model and overland flooding model using
Delft3D Flexible Mesh (DFM), illustrating the serious consequences from the perspective of
storm tide. Based on the probability distribution of storm tide, this study regards TC1415 as the
100-year event, TC6311 as the 50-year event, TC8616 as the 25-year event, TC8007 as the
10-year event, and TC7109 as the 5-year event. The results indicate that the coastal area is a major
floodplain, primarily due to storm tide, with the inundation severity positively correlated with the
height of the storm tide. For 100-year TC event, the inundation area with a depth above 1.0 m
increases by approximately 2.5 times when compared with 5-year TC event. The comparison of
single-driven flood (storm tide flooding and rainstorm inundation) and compound flood hazards
shows that simply accumulating every single-driven flood hazard to define the compound flood
hazard may cause underestimation. For future research on compound flooding, copula function
can be adopted to investigate the joint occurrence of storm tide and rainstorm to reveal the severity
of extreme TC flood hazards.
**Keywords** Tropical cyclones; Compound flooding; Storm tide; Rainstorm; Coastal cities


## 1 Introduction

Flood hazards, especially those happening during tropical cyclones (TCs), have become the most devasting and expensive natural hazards of coastal cities (Patricola and Wehner, 2018; van Oldenborgh et al., 2017; Hallegatte et al., 2013; Adelekan, 2011). Storm tides brought on by TCs can lead to coastal flooding, and rainstorms occurring during TCs can lead to urban inundation. The simultaneous or consecutive occurrence of storm tide and rainstorm in time and/or space can lead to compound flooding (Gori et al., 2020b; Wahl et al., 2015; Leonard et al., 2014). In the past decade, many compound flood hazards occurred in coastal regions worldwide due to TCs, such as Typhoon Irma (2017) in Jacksonville and Typhoon Lekima on China's southeast coast. An extremely destructive flood event in Houston-Galveston during Hurricane Harvey (2017) was confirmed to be a compound flood hazard (Huang et al., 2021). It was caused by land-derived runoff (mainly considered to be rainfall) and ocean-derived forcing (mainly considered to be storm tide) (Valle-Levinson et al., 2020). The coastal region suffered a major economic loss of more than 125 billion dollars from Harvey. Thus, it is important to investigate the compound flood risk during TCs to better comprehend flood hazards in coastal cities.

The projection of future climate change indicates that TCs will occur more frequently with greater intensity. Accordingly, the likelihood of the co-occurrence of storm tide and rainstorm will increase drastically (Keellings and Hernández Ayala, 2019; Marsooli et al., 2019; Emanuel, 2017; Lin et al., 2012), which may cause more extreme compound flood hazards (Bevacqua et al., 2019; Rasmussen et al., 2017; Wahl et al., 2015; Milly et al., 2002). Due to global warming, sea level rise, land subsidence, and urban expansion, coastal cities are confronted with the critical threat of TC compound flooding (Yin et al., 2021, 2020; Wang and Tan, 2021; Hsiao et al., 2021; Wang et al., 2018). Recent studies evaluated compound flood risk at the regional scale (Fang et al., 2020; Bevacqua et al., 2019; Hendry et al., 2019; Budiyono et al., 2016; Wahl et al., 2015). Wahl et al. (2015) assessed the risk of compound flooding from rainfall and storm surge in major US cities. Bevacqua et al. (2019) estimated the probability of compound flooding from precipitation and storm surge in Europe. Both studies showed that there will be an increase of compound flood risk in coastal cities in the future. A study conducted by Fang et al. (2020) investigated the compound flood potential from precipitation and storm surge in coastal China, indicating that low-latitude (<30°N) coastal areas in southeast China are more prone to compound flood hazards from storm tide and rainfall during TCs.

Only several urban-scale studies on compound flooding have been carried out in China (He et al., 2020; Wang et al., 2019; Xu et al., 2018; Yin et al., 2016). Lian et al. (2013) investigated the joint impact of rainfall and tidal level on flood risk in Fuzhou City. Xu et al. (2014) analyzed the joint probability of rainfall and storm tide under changing environment, concluding that the probability



of compound flooding would increase by more than 300% in Fuzhou. Lian et al. (2017) identified
the major hazard-causing factors of compound flooding and classified the floodplains into tidal
zone, hydrological zone, and transition zone in Haikou City. Although studies such as these have
investigated the joint risk of hazard-causing factors in compound floods, they seldom pay attention
to the compound flooding that occurs during TCs.

Most studies concerned with compound flooding rely on historical data, which contains
information on hourly storm tide and daily rainfall (Yum et al., 2021; Fang et al., 2020; Zellou and
Rahali, 2019; Wu et al., 2018; Lian et al., 2017). The recorded data is often used to investigate the
statistical correlation between flood drivers (Xu et al., 2019, 2014; Xu et al., 2018; Lian et al.,
2013). Based on the recorded storm tide from 49 tide gauges and daily precipitation from 4890
rainfall stations in Australia, Zheng et al. (2013) quantified the dependence between rainfall and
storm surge to investigate flood risk in coastal zones. However, for a number of coastal regions in
the world, it is difficult to obtain sufficient recorded data that can be used to analyze the
mechanism of TC compound flooding from storm tide and rainfall. An alternative approach is
applying a hydrodynamic model to simulate storm tides (Gori et al., 2020a). For example, Yin et
al. (2021) constructed a storm surge model to simulate the storm tide derived from 5000 synthetic
TCs for the estimation of TC-induced coastal flood inundation.

Hydrodynamic models can also be employed to simulate flood events (Bevacqua et al., 2019;
Zellou and Rahali, 2019; Kumbier et al., 2018). It is an effective method to model the flood extent
and inundation depth, this method has generally been applied in research on single-driven flood
hazard (Wang et al., 2018, 2012; Yin et al., 2013). Recently, many studies have used
hydrodynamic models to simulate compound flood events driven by historical TC events or
synthetic TC scenarios (Bilskie et al., 2021; Orton et al., 2020; Santiago-Collazo et al., 2019; Shen
et al., 2019). Gori et al. (2020b) constructed a coupled framework of three models to simulate
storm surges and compound flood events. This method has the advantage of observing the
spatiotemporal dynamics of rainfall and storm surges during TCs (Gori et al., 2020b; Orton et al.,
2020). However, assessing the compound flood risk by constructing a coupled model is not
commonly used in current studies on compound flood hazards, mainly because the simulation of
compound flooding involves multiple driving condition settings and requires combining multiple
physics-based models.

Delft3D Flexible Mesh (DFM), developed by Deltares, Netherland, has been widely applied to
build storm surge numerical models for research on storm surge because of its capability of
simulating 2D and 3D shallow water flow (De Goede, 2020). It integrates Delft3D-FLOW model
suites and uses flexible unstructured grids, which is convenient for partial grid refinement
(Deltares, 2018). A recent study on compound flooding utilized this model to simulate storm



surges for characterizing extreme sea level, investigating the probability of compound floods from
precipitation and storm surge in Europe (Bevacqua et al., 2019). Meijer and Hutten (2018)
developed a 2D urban model with DFM for the downtown area of Shanghai. The results indicated
that DFM was capable of modeling rainfall-runoff and could be used to construct urban flood
models. Therefore, it is feasible to simulate both storm surge and rainfall-runoff based on DFM to
assess compound flooding.
This study investigates the compound effect of flooding from storm tide and rainstorm during TCs
to better understand of compound flooding in Haikou. Based on the DFM model, we set up a
storm surge model and overland flooding model to simulate the floodplain under TC events. We
select 66 TC events that influenced Haikou to explore the probability distribution of storm tide,
further selecting 5 TC events that respectively corresponds to the 5-, 10-, 25-, 50-, and 100-year
return period. The risk of rainstorm inundation, storm tide flooding, and compound flooding are
quantitively assessed and compared based on the simulation results under different return periods.
The conclusions drawn from this study can provide insight into mitigating compound flood risk in
coastal areas.
To the best of our knowledge, this is the first study that applies a coupled model by DFM to assess
TC compound flood risk in Haikou. The objectives of this study include (1) investigating the
probability of storm tide during TCs by modelling TCs influenced Haikou; (2) quantifying the
compound effects of rainfall and storm surge under TC events of different return periods; (3)
assessing and comparing the flood severity of rainstorm inundation, storm tide flooding, and
compound flooding.
This paper is organized as follows: Section 2 presents the background information about the study
area and data requirements. Section 3 describes the model configuration and explains how TCs
that influenced Haikou were selected. The method of how to assess the compound flood risk is
also in this section. Model verification and the analysis of probability distribution of storm tide are
reported and discussed in Section 4. The assessment and comparison of rainstorm inundation,
storm tide flooding, and compound flooding are also discussed in this section. Finally, conclusions
are given in Section 5.
**2 Materials**
**2.1 Study area**
Haikou is located in the north of Hainan Island, China, where the geographical position is
relatively independent (Figure 1). The coastal area of Haikou is low and flat. In particular, the
elevation of downstream plain and areas along Nandu River (in Figure 1) is less than 3.0m.
Haikou is frequently affected by TCs and rainstorms from June to October, the annual rainfall is
around 1660 mm. Storm tide flooding caused by TCs is one of the main natural hazards in Haikou,
roughly 3 storm surges have occurred in Haikou every year in recent decades. The combination of
storm tide and rainstorm will increase the probability of extreme compound flooding, posing a
threat to social infrastructure and urban traffic in Haikou. During Typhoon Kalmaegi (2014), a
total of 219.8 mm (24h) of precipitation were produced and the highest tide level reached 4.3 m in
Haikou. The occurrence of heavy rainfall and strong storm tide caused serious compound flooding
with a 220-million-dollar economic loss. Under the changing environment, Haikou will face
greater compound flooding risks and challenges from TCs, storm surges, and rainstorms in the
future.

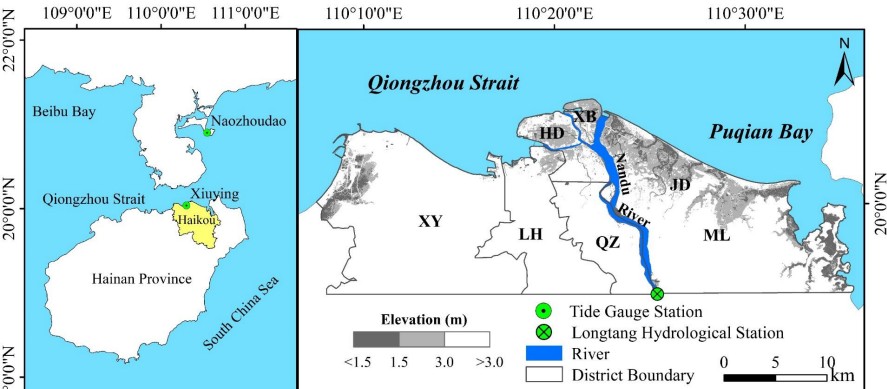

Figure 1. The geographic location of tide stations and Nandu river in Haikou, and the basic
geographic information of Haikou (XY: Xiuying district, LH: Longhua district, QZ: Qiongzhou
district, ML: Meilan district, HD: Haidian Island, XB: Xinbu Island, JD: Jiangdong New Area).
**2.2 Data**
The geographic and meteorological data of the study area were systematically collected in this
study (Table 1). The topographic map of the study area was provided by Hainan Emergency
Management Department, and the bathymetry data of South China Sea and Beibu Bay was
obtained from General Bathymetric Chart of the Oceans (GEBCO). The spatial resolution of the
topographic map is 5 m, and the bathymetry data is 500 m. The meteorological data includes
historical TC track data and daily rainfall data from 1960 to 2017. The historical TC track data
containing the TCs location (latitude and longitude), two-minute mean maximum sustained wind
(*MSW; m/s*), and minimum pressure (*hPa*) near the TC center, was provided by Shanghai Typhoon
Institute of China Meteorological Administration. The daily rainfall data of Haikou was
downloaded from the CMA website (http://data.cma.cn/), and can be transferred to hourly rainfall
by interpolation for inundation simulation (Ye et al., 2018; Yang et al., 2013). The annual river





discharges at Longtang hydrological station from 1960 to 2020 were provided by Hainan
Hydrology and Water Resources Survey Bureau.

Table 1. Data profile of this study

| Type | Name | Attributes | Source |
|------|------|------------|--------|
| Basic data | DEM, Haikou | 2018, 5m | Department of Emergency Management of Hainan Province |
|  | DEM, bathymetry | 2019, 500m | https://www.gebco.net |
| Meteorological data | TC tracks | 1949-2019, 3 hourly | Shanghai Typhoon Institute of China Meteorological Administration |
|  | Rainfall | 1960-2017, daily | http://data.cma.cn |
|  | Discharge | 1960-2020, daily | Hainan Hydrology and Water Resources Survey Bureau |


**3 Methods**
**3.1 Model configuration and validation methods**
Delft3D Flexible Mesh (DFM) developed by Deltares in 2011 is a practical unstructured shallow
water flow calculation model (De Goede, 2020). It can be used for both ocean hydrodynamic and
surface runoff numerical simulations (Kumbier et al., 2018; Meijer and Hutten, 2018). In this
study, the DFM model was established to calculate the hydraulic boundary conditions needed to
estimate overland flow boundary, and simulate the overland inundation during TCs period (Gori et
al., 2020b).
**3.1.1 Storm surge model**

The calculation domain of the storm surge model covers Hainan Province, the South East Sea, and
Beibu Bay, and roughly ranges from 15 to 24.5°N and 105.5 to 118.5°E (Figure 1). The minimum
mesh grid size is 100 m and the maximum mesh grid size is 12000 m. Astronomical tide is
simulated by importing the phase and amplitude of tidal constituents (Q1, P1, O1, K1, N2, M2, S2,
and K2) extracted from the global tidal model (TPXO8.0). A built-in module in Delft3D WES
(Wind Enhance Scheme) module is employed to calculate the TCs wind field according to Holland
formula (Holland, 1980). We use the statistical measures *RMSE (Root Mean Square Error)* and $R^2$
to evaluate the model performance of simulated tide (Kumbier et al., 2018; Skinner et al., 2015).
The storm surge model is validated against measured astronomical tide and storm tide
(astronomical tide plus storm surge). Storm tide series (TC1415, "Kalmaegi") at Xiuying gauge
station were collected from Haikou Municipal Water Authority to validate this model. For the
validation of astronomical tide, we also collected astronomical tide for TC1415 from Xiuying and
Naozhoudao tide gauge station. All tide levels were recorded every hour (from 00:00 on
September 15, 2014 to 00:00 on September 17, 2014).





### 3.1.2 Overland flooding model

The overland flooding model combines regular and irregular triangular mesh. This model is a
surface runoff numerical model and the mesh grid resolution is set as 50 m. The average annual
discharge (165.81 m³/s) at Longtang hydrological station is calculated as the upstream boundary
condition. In this model, the storm tide series extracted from the storm surge model serve as the
coastal boundary conditions. This model is validated against the measured inundation area and
depth. We collect the inundation data of TC1415 and conduct a fieldwork of Haikou for the
validation of this model. By comparing the inundation map of TC1415 with measured inundation
area and depth, the overland inundation model can be approximately validated.

### 3.2 TCs influencing Haikou


The TCs that pass through the region (18-22°N, 109-113°E) and stay over 24 hours have an
apparent effect on Haikou (Ding, 1999; Wang, 1998; He, 1988). 66 TCs from 1960 to 2017 are
selected in this study (Figure 2), and we construct typhoon wind fields and simulate the storm tide
of these TCs. Each TC event has a code, for example, the ninth typhoon in 1963 is coded as
TC6309.

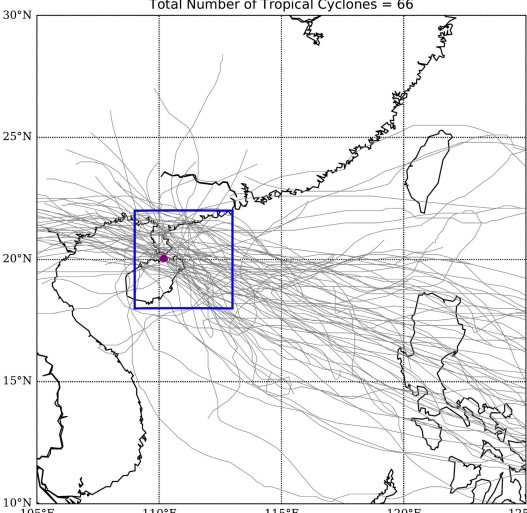


Figure 2. Location map for the study area. Purple dot indicates the location of Haikou. Grey
colored lines indicate major historical TC tracks within the region. Blue box indicates the
selection region (18-22°N, 109-113°E)

### 3.3 Compound flooding assessment



In this study, we investigate the probability distribution of storm tides to assess compound flood
hazards. Based on the storm surge model, the storm tide series of 66 TCs are simulated. The
highest storm tides during TCs are used to calculate the probability distribution function at
Xiuying tide gauge station.

Exploring the storm tide distribution can offer comprehension of the probability of compound
flood hazards from storm surge. Extreme Value Distribution (EVD) is widely applied to
investigate storm tide probability distribution (Yum et al., 2021). We assume that the storm tide
fits either Gumbel or Weibull extreme value functions, then calculate their function fitting
parameters. We compare the goodness-of-fit of two distribution functions (Gumbel, Weibull) with
Kolmogorov-Smirnov (K-S) test. K-S test is an appropriate method to explore the distribution of
continuous random variables, and can be used to select the best fitting distribution function.
According to the storm tide distribution, we can achieve tide levels at different probabilities (P).
We replace P with storm tide return periods (T), which equals to 1/P, to investigate the possibility
of extreme storm tide. The corresponding TC events in 5-, 10-, 25-, 50-, and 100-year return
period can be found to compare the compound flood hazards with different storm tides.

**4 Results and discussion**
**4.1 Model validation**

We use TC1415 to verify the astronomical tide and storm tide of the storm surge model. In the
validation of astronomical tide, we use the predicted astronomical tide at two gauge stations;
Naozhoudao (Zhanjiang, Guangdong) and Xiuying (Haikou, Hainan). The calculation results
show that the $RMSE$ is 0.18 m and 0.14 m for Naozhoudao and Xiuying gauge station, the $R^2$ of
both Naozhoudao and Xiuying gauge station are 0.91. Figure 3(a) and (b) depict simulated and
predicted water level at Xiuying and Naozhoudao gauge station. The curves of simulated
astronomical tide at the two stations fit observed tide level points well. Thus, this model has a
good ability to simulate astronomical tides. In the validation of storm tide, we add the wind field
of TC1415 in the model and only use the observed tide level at Xiuying gauge station for
validation (Figure 3(c)). The calculation of $RMSE$ is 0.34 and $R^2$ is 0.83. It can be seen from
Figure 3(c) that the curve of simulated storm tide is consistent with the observation, and the
highest storm tide is well simulated.


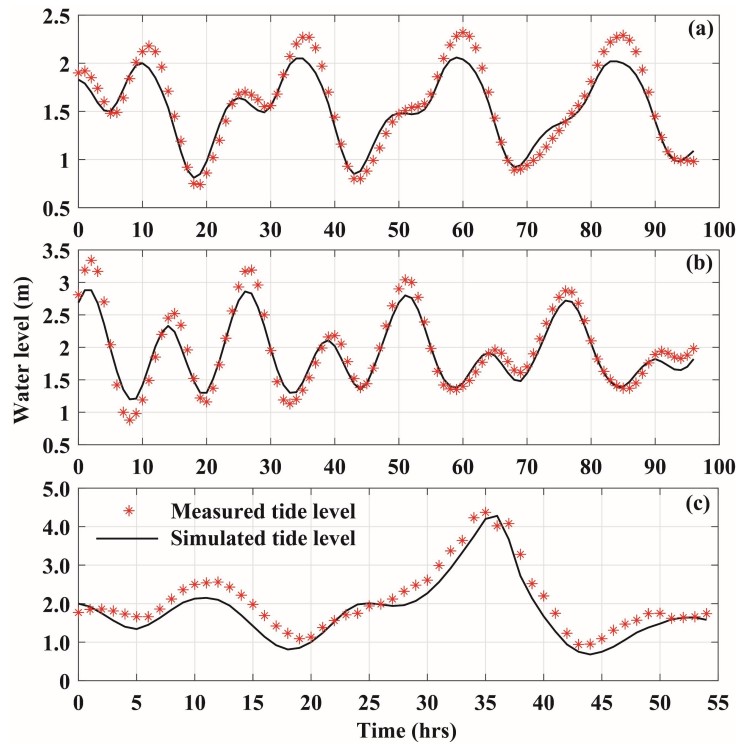


Figure 3. The simulation results of astronomical tide and storm tide compared to measured tide levels. (a. astronomical tide at Xiuying gauge station. b. astronomical tide at Naozhoudao gauge station. c. storm tide at Xiuying gauge station. Black lines indicate the simulated tide level, red asterisk points indicate measured tide level.)


Tide levels along the coastline extracted from the storm surge model serve as coastal boundary
conditions for the overland flooding model. We utilize the TC1415 event to also validate the
overland flooding model. Comparing the simulation of compound flooding with the measured
inundation of roads during TC1415 (Kuang, 2014), the main inundation area in the simulation is
coincident with the flooded roads (Figure 4). Furthermore, the distribution of simulated inundation
area is also consistent with the actual flood distribution, hence this overland flooding model has
good capacity of modelling and demonstrating TC flood hazards.



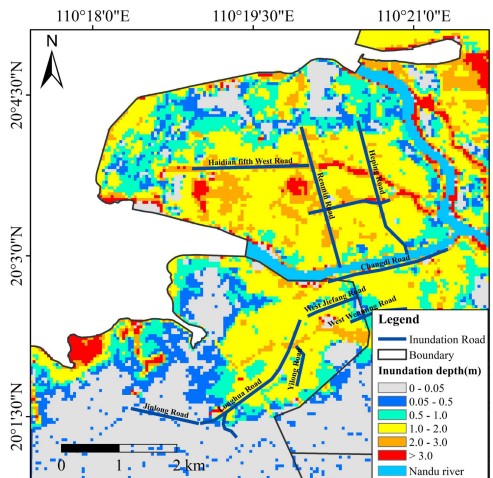


Figure 4. Spatial extent of simulated and measured inundation area and depth during TC1415.


**4.2 Storm tide probability distribution**


Xiuying gauge station is selected as a representative location to examine the probability
exceedance of TC storm tide. Storm tide return period is calculated based on the maximum storm
tide in the past 58 years simulated for 66 TCs. The results of K-S test show that the D-value and
P-value of GUM are 0.0615 and 0.9995, while the D-value and P-value of WEI are 0.0769 and
0.9876. Thus, the Gumbel extreme value (GEV) distribution function can fit TC storm tide better.
Figure 5 shows that GEV fits storm tide well, presenting the corresponding TCs under different
return periods. Red circles represent the maximum storm tide from the 66 TCs in the past. The
solid line represents estimation of the GEV fitting.

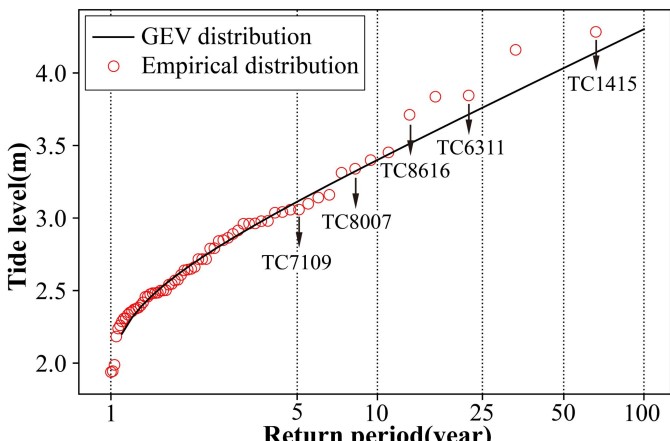


Figure 5. Storm tide at Xiuying gauge station as a function of return period based on GEV fitting
(solid line).






Table 2 shows the corresponding TC events and their highest storm tide and accumulated rainfall
under different return periods. TC1415, with the highest storm tide, is considered a 100-year event.
In order to investigate the compound effects of storm tide and rainstorm on the overland
inundation, TCs with higher accumulated rainfall are selected. As a result, TC6311, TC8616,
TC8007, and TC7109 are assigned to 50, 25, 10, and 5 years based on GEV fitting, respectively.

Table 2. The different return periods of TC storm tide and the related TC events.

| Return period | Event | Water level (m) | Rainfall (mm) |
| --- | --- | --- | --- |
| 5Y | TC7109 | 3.04 | 137.7 |
| 10Y | TC8007 | 3.31 | 196.0 |
| 25Y | TC8616 | 3.71 | 128.0 |
| 50Y | TC6311 | 3.84 | 191.0 |
| 100Y | TC1415 | 4.28 | 219.8 |


**4.3 Compound flooding assessment in different storm tide return periods**

Figure 6 presents the compound flood inundation maps under 5-, 10-, 25-, 50-, and 100-year
return period. For 5-year inundation map, the major inundation area is distributed along the
Jiangdong New Area and Xinbu Island on the northeast coast. The inundation area with sporadic
distribution is caused by rainfall in the inland urban area. As return periods increase, Haidian
Island, north Longhua district and northwest Xiuying district begin to have serious flood extents,
and the compound flooding severity of Jiangdong New Area and Xinbu Island increases. For
100-year return period, the inundation depth regions are above 1.0 m, and the floodplain depth is
above 3.0 m in most of Jiangdong New Area. Regions with inundation depth below 0.05 m are not
evaluated in this study due to their low risk.




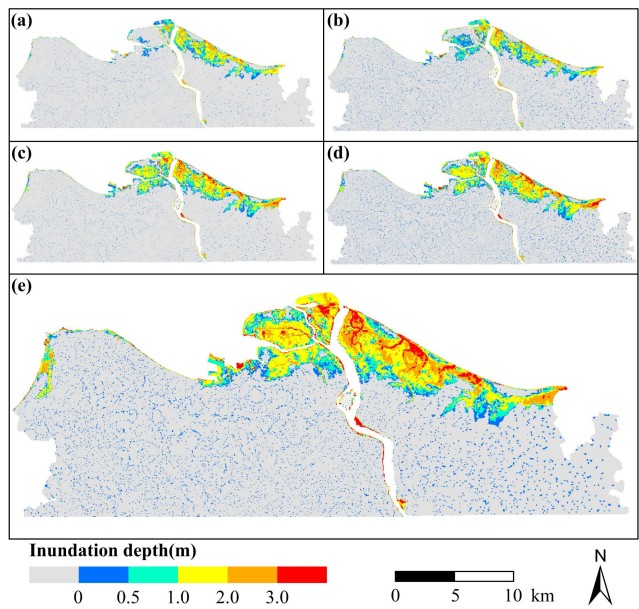


Figure 6. The compound flood inundation maps under different return period (a. 5-year event, b.
10-year event, c. 25-year event, d. 50-year event, and e. 100-year event).

Table 3 indicates the inundation depth and area under different return periods. In 100-year TC
event, the total inundation area is 12613 ha, and the inundation area between 0-0.5 m and 1.0-2.0
m accounts for 29.4% and 31.1%, respectively. The inundation area between 0.5-1.0 m and 2.0-3.0
m accounts for a total of 32.7%. For the other TC events, the inundation depth at a range of 0-0.5
m and 1.0-2.0 m has the most inundation area. For 100-year TC event, the inundation area with a
depth above 1.0 m increases by approximately 2.5 times when compared with 5-year TC event.

Table 3. Inundation depth and area under different return periods.

| Flooding depth(m) | Flooding area(ha) | | | | |
|---|---|---|---|---|---|
| | 5-year | 10-year | 25-year | 50-year | 100-year |
| 0 – 0.5 | 2139 | 3757 | 2364 | 3957 | 3704 |
| 0.5 – 1.0 | 1349 | 1623 | 2037 | 1965 | 2065 |
| 1.0 – 2.0 | 1884 | 1980 | 3035 | 3513 | 3927 |
| 2.0 – 3.0 | 818 | 879 | 1389 | 1511 | 2055 |
| >3.0 | 29 | 112 | 384 | 516 | 862 |
| Total | 6219 | 8351 | 9209 | 11462 | 12613 |


**4.4 Quantitative comparison single-driven flood hazard and compound flood**
**hazard**

Figure 7 illustrates the maps of rainstorm inundation and storm tide flooding under different return
periods. In each rainfall scenario, the overall inundation depth is below 1.0 m, while in each storm





tide scenario, the overall inundation depth is above 1.0 m. When comparing the rainstorm
inundation map and storm tide flooding map in the same TC event, it is obvious that the storm tide
flooding is significantly worse than rainstorm inundation.

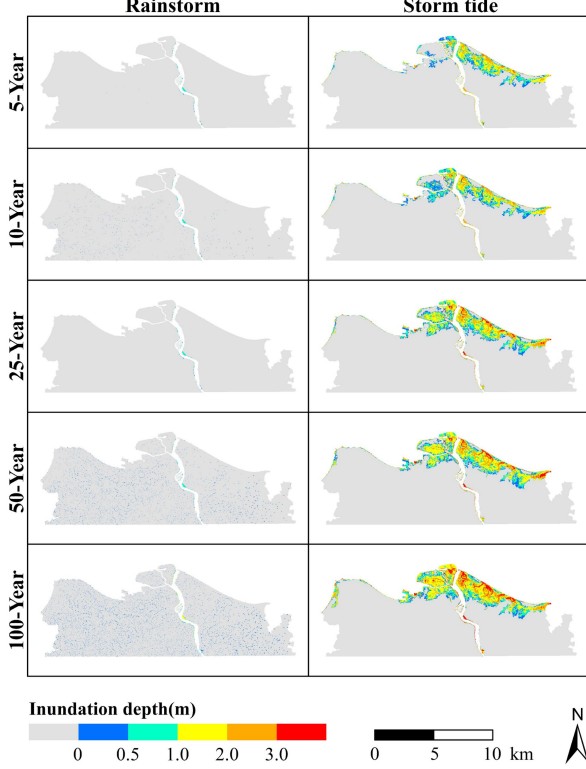


Figure 7. The inundation maps of rainstorm and storm flooding under different return periods.


Figure 8 shows the comparison of overall inundation area of rainstorm, storm tide, and compound
flooding under different return periods. The inundation area of compound flooding exceeds the
inundation area of rainstorm inundation and storm tide flooding in each TC event. Thus,
compound flood hazards can have more serious consequences than rainstorm and storm flooding
(Bevacqua et al., 2019; Wahl et al., 2015a; Zscheischler et al., 2018). Moreover, it can be seen
from Figure 8 that compound flooding has more inundation area than the accumulation of
rainstorm and storm tide flooding under different return periods. For example, in the TC6311
scenario, the total inundation area of compound flooding is 11462 ha, exceeding the sum of
rainstorm inundation and storm tide flooding, which is 10616 ha. Therefore, compound flood
hazards are more destructive than the combination of single-driven flood hazards, and have a
certain amplification effect (Fang et al., 2020; Xu et al., 2019).





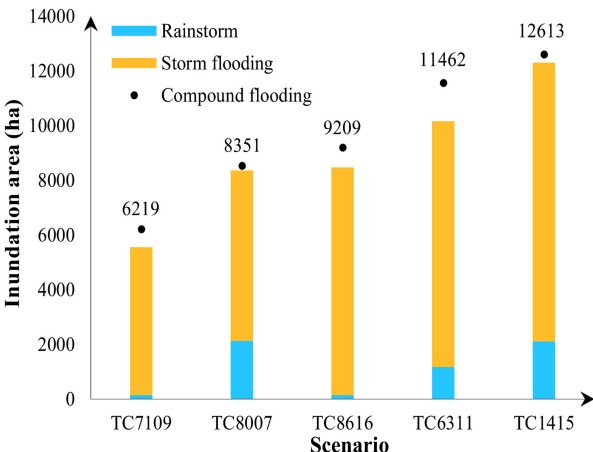


Figure 8. The comparison of the overall inundation area of rainstorm, storm flooding, and

compound flooding in each TC event.


However, storm tide and rainstorm are the driving factors in a compound flood hazard (Hsiao et
al., 2021; Fang et al., 2020; Bevacqua et al., 2019). In this study, we investigate the compound
effect of flood hazards by studying the probability distribution of highest storm tides during TCs.
Many studies have confirmed that rainfall and storm surge have statistically positive dependence
(Wahl et al., 2015; Xu et al., 2014; Zheng et al., 2014). Hence, it is of practical significance to
reveal compound flood risk considering the statistical dependence of rainfall and storm surge.
Copula function has been confirmed that to not only model and describe the dependence between
flood variables, but also to express compound flood risk (Zellou and Rahali, 2019; Xu et al., 2019;
Wu et al., 2018; Lian et al., 2013b). In future works, we will adopt the copula function to
investigate the joint occurrence of rainfall and storm surge during TCs, further assessing extreme
compound flooding severity.

**5 Conclusions**

This study applies a coupled methodology of combining storm surge model and overland flooding
model to investigate the compound effect of flood hazards during TCs. We simulate and assess
compound floods under different return periods of storm tides. The results show that storm tide is
the key driving factor of compound flood inundation in Haikou, and tide level decides the
inundation extent. When quantitively comparing compound flooding with rainstorm inundation
and storm flooding, we find that it is more destructive than single-driven flood hazards, and the
compound effect exceeds the accumulated effects of single-driven floods. The co-occurrence of
heavy rainfall and strong storm surge in extreme TCs could intensify compound flood inundation.
Simply accumulating every single-driven flood hazard to define compound flooding may cause


underestimation.

Although this study is limited to Haikou City, the methodology of quantitatively assessing
compound flooding risks through constructing a coupled framework of two hydrodynamic models
is available for other coastal cities. It can conveniently capture the dynamic of rainfall and storm
surge, and directly observe the change of inundation area to display the effect of rainfall and storm
surge in compound events. For future research on extreme TC compound flooding, copula
function can be applied to study the statistical dependence between heavy rainfall and strong
storm surge, revealing extreme flood risk in coastal cities.

**Data availability**
Some of the used data such as the typhoon tracks in this study are freely available. The web links
are presented in Section 2. However, some data such as the topology map of the study area and
river discharges were provided on the request from the departments and agencies of Haikou.

**Author contribution**
QL, HQX and JW designed the study. QL construct and validated the models, and ran all the
simulations. QL and HQX analyzed and interpreted the results. QL wrote, reviewed, and edited the
manuscript. HQX and JW reviewed the manuscript.

**Competing interests**

The authors declare that they have no conflict of financial interest that could have appeared to
influence the work reported in this paper.

**Acknowledgements**

The authors express gratitude to the Department of Emergency Management of Hainan Province
Hainan Hydrology and Water Resources Survey Bureau for supporting the geographic information
of the study area. This work was supported by the National Key Research and Development
Program of China (2018YFC1508803), and the National Social Science Foundation of China
(18ZDA105).





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
