# Peer review of "Assessing tropical cyclone compound flood risk using hydrodynamic"

_Natural Hazards and Earth System Sciences, 2021_

## Author Response (AR1)

**Overall Response**: We would like to thank the associate editor and two reviewers for their detailed comments and suggestions, which are very helpful for us to improve the quality of the manuscript. In the revision, we have taken into consideration all suggestions and addressed all concerns raised. As a consequence, the quality of the manuscript increased and we hope that it is acceptable. In this letter we report the point-by-point response to the comments of the two reviewers.

The authors present an interesting study on the effect of tropical cyclone compound flood to Haikou City using hydrodynamic modelling. The paper is well written and organized. I believe the paper will be publishable with some minor revisions and additions.

In more general terms:

1. The English needs some proof reading, please refer to the specific term below.
We have reorganized the manuscript and carefully revised each section. The results and conclusion sections were largely re-written.

2. The authors entitle the manuscript "Assessing tropical cyclone compound flood risk using hydrodynamic modelling: a case study in Haikou City, China", emphasizing the method of "hydrodynamic modelling", then the advantage and disadvantage of the method should be discussed.
We re-wrote the method section to explain the advantage and disadvantage of the hydrodynamic modelling method:

Line 92: This method has the advantage of observing the spatiotemporal dynamics of rainfall and storm surges during TCs (Gori et al., 2020b; Orton et al., 2020). However, assessing the compound flood risk by constructing a coupled model is not commonly used in current studies on compound flood hazards, mainly because the simulation of compound flooding involves multiple driving condition settings and requires combining multiple physics-based models.

References:
Gori, A., Lin, N., Xi, D.: Tropical cyclone compound flood hazard assessment: from investigating drivers to quantifying extreme water levels. Earths Future, 8, https://doi.org/10.1029/2020EF001660, 2020b.
Orton, P., Conticello, F., Cioffi, F., et al.: Flood hazard assessment from storm tides, rain and sea level rise for a tidal river estuary. Nat. Hazards, 102, 729–757, https://doi.org/10.1007/s11069-018-3251-x, 2020.

3. The authors do not discuss the function of seawall – which is crucial on the extent of flooding area caused by sea hazard. I recommend some discussion of this topic in the paper.

It is because we are interested in compound flooding effect. To cope with the influence from seawall. We collected the information of the construction standard of seawall and compared it with the high-resolution (5 m) DEM of Haikou, finding that the effect of seawall can be reflected by the DEM roughly. In the future, we would import more accurate seawall data in the overland flooding model. We have included a discussion in the paper.:

Line 202: The high-resolution topography of study area is imported in the model, and it can roughly reflect the effect of seawall.

4. The authors should add to the literature review and discussions more detail previous literature looking at the copula function of compound flooding (e.g. Lin-Ye, et al., 2016; Zhang et al., 2021), since it is the future research of your work.

Min Zhang; Zhijun Dai; Tjeerd J Bouma; Jeremy Bricker; Ian Townend; Jiahong Wen; Tongtiegang Zhao; Huayang Cai; Tidal-flat reclamation aggravates potential risk from storm impacts, Coastal Engineering, 2021, 166: 103868.

Lin-Ye, J., Garcia-Leon, M., Gracia, V. and Sanchez-Arcilla, A., 2016. A multivariate statistical model of extreme events: an application to the Catalan coast. Coastal Engineering 117, 138-156.

Thanks for pointing to these papers. We rephrased the sentence as follows:

Line 355: Copula is a kind of function connecting joint distributions and marginal distributions (Lin-Ye et al., 2016). Zhang et al. (2021) calculated the overtopping occurrence by determining the correlations between tidal levels and wave heights based on copula function. In recent years, copula function has been confirmed to model and describe the dependence between flood variables and express compound flood risk (Zellou and Rahali, 2019). Xu et al. (2019) employed copula function to investigate the bivariate return period of compounding rainfall and storm tide events, finding that joint probability analysis can reveal more adequate and comprehensive risk about compound events than univariate analysis.

References:
Lin-Ye, J., Garcia-Leon, M., Gracia, V. and Sanchez-Arcilla, A., 2016. A multivariate statistical model of extreme events: an application to the Catalan coast. Coastal Engineering 117, 138-156.
Zhang, M., Dai, Z., Bouma, J., et al.: Tidal-flat reclamation aggravates potential risk from storm impacts, Coastal Engineering, 166: 103868. https://doi.org/10.1016/j.coastaleng.2021.103868, 2021.
Zellou, B., Rahali, H. Assessment of the joint impact of extreme rainfall and storm surge on the risk of flooding in a coastal area. J. Hydrol., 569, 647–665, https://doi.org/10.1016/j.jhydrol.2018.12.028, 2019.
Xu, H., Xu, K., Lian, J., et al.: Compound effects of rainfall and storm tides on coastal flooding risk. Stoch. Environ. Res. Risk Assess, 33, 1249–1261, https://doi.org/10.1007/s00477-019-01695-x, 2019.

In more specific terms:

19-21->For a 100-year TC event, the inundation area with a depth above 1.0 m increases by approximately 2.5 times compared with a 5-year TC event. Comparing the single-driven flood (storm tide flooding and rainstorm inundation) and compound flood hazards
Line19-21, we rephrased the sentence as follows:

For 100-year TC event, the inundation area with a depth above 1.0 m increases by approximately 2.5 times when compared with 5-year TC event. Comparing single-driven flood (storm tide flooding and rainstorm inundation) and compound flood hazards shows that simply accumulating every single-driven flood hazard to define the compound flood hazard may cause underestimation.

23->For future research on compound flooding, the copula function
Line23, we rephrased the sentence as follows:

For future research on compound flooding, the copula function can be adopted to investigate the joint occurrence of storm tide and rainstorm to reveal the severity of extreme TC flood hazards.

42-43-> Thus, it is important to investigate the compound flood risk during TCs to comprehend flood hazards in coastal cities better.
Line42-43, we rephrased the sentence as follows:

Thus, it is important to investigate the compound flood risk during TCs to comprehend flood hazards in coastal cities better.

49-> Due to global warming, sea-level rise, land
Line49, we rephrased the sentence as follows:

Due to global warming, sea-level rise, land subsidence, and urban expansion,

56-> Both studies showed that there would be an increase in compound flood risk in coastal cities in the future

Line56, we rephrased the sentence as follows:

Both studies showed that there would be an increase in compound flood risk in coastal cities in the future.

66-68-> Lian et al. (2017) identified the major hazard-causing factors of compound flooding and classified the floodplains into tidal, hydrological, and transition zones in Haikou City.

Line66, we rephrased the sentence as follows:

Lian et al. (2017) identified the major hazard-causing factors of compound flooding and classified the floodplains into tidal, hydrological, and transition zones in Haikou City.

Reference:
Lian, J., Xu, H., Xu, K., et al.: Optimal management of the flooding risk caused by the joint occurrence of extreme rainfall and high tide level in a coastal city. Nat. Hazards, 89, 183–200, https://doi.org/10.1007/s11069-017-2958-4, 2017.

75-77-> For example, based on the recorded storm tide from 49 tide gauges and daily precipitation from 4890 rainfall stations in Australia

Line75, we rephrased the sentence as follows:

For example, based on the recorded storm tide from 49 tide gauges and daily precipitation from 4890 rainfall stations in Australia,

78-79-> However, for many coastal regions in the world, it is difficult to obtain

Line78, we rephrased the sentence as follows:

However, for many coastal regions in the world, it is difficult to obtain sufficient recorded data that can be used to analyze the mechanism of TC compound flooding from storm tide and rainfall.

81-83-> For example, Yin et al. (2021) constructed a storm surge model to simulate the storm tide derived from 5000 synthetic TCs to estimate TC-induced coastal flood inundation.

Line81, we rephrased the sentence as follows:

For example, Yin et al. (2021) constructed a storm surge model to simulate the storm tide derived from 5000 synthetic TCs to estimate TC-induced coastal flood inundation.

Reference:
Yin, J., Lin, N., Yang, Y., et al.: Hazard Assessment for Typhoon‐Induced Coastal Flooding and Inundation in Shanghai, China. J. Geophys. Res. Oceans, 126, https://doi.org/10.1029/2021JC017319, 2021.

86-88-> It is an effective method to model the flood extent and inundation depth, and this method has generally been applied in research on single-driven flood hazards

Line86, we rephrased the sentence as follows:

It is an effective method to model the flood extent and inundation depth, and this method has generally been applied in research on single-driven flood hazards

98-99-> has been widely applied to build storm-surge numerical models

Line98, we rephrased the sentence as follows:

Delft3D Flexible Mesh (DFM), developed by Deltares, Netherland, has been widely applied to build storm-surge numerical models for research on storm surge because of its capability of simulating

2D and 3D shallow water flow

100-101-> It integrates Delft3D-FLOW model suites and uses flexible unstructured grids, convenient for partial
Line100, we rephrased the sentence as follows:

It integrates Delft3D-FLOW model suites and uses flexible unstructured grids, convenient for partial grid refinement

103-> characterizing extreme sea levels,
Line101, we rephrased the sentence as follows:

A recent study on compound flooding utilized this model to simulate storm surges for characterizing extreme sea levels,

107-> Therefore, it is feasible to simulate storm surge and rainfall-runoff based on DFM to assess compound flooding.
Line107, we rephrased the sentence as follows:

Therefore, it is feasible to simulate storm surge and rainfall-runoff based on DFM to assess compound flooding.

110-114-> This study investigates the compound effect of flooding from storm tide and rainstorm during TCs to understand compound flooding in Haikou better. We set up a storm surge model and overland flooding model based on the DFM model to simulate the floodplain under TC events.
Line109, we rephrased the sentence as follows:

This study investigates the compound effect of flooding from storm tide and rainstorm during TCs to understand compound flooding in Haikou better. We set up a storm surge model and overland flooding model based on the DFM model to simulate the floodplain under TC events.

115-116-> further selecting 5 TC events that correspond to the 5-, 10-, 25-, 50-, and 100-year return periods, respectively.
Line112, we rephrased the sentence as follows:

We select 66 TC events that influenced Haikou to explore the probability distribution of storm tide, further selecting 5 TC events that correspond to the 5-, 10-, 25-, 50-, and 100-year return periods, respectively.

142-> rainstorms from June to October. The annual rainfall is around 1660 mm.
Line139, we rephrased the sentence as follows:

Haikou is frequently affected by TCs and rainstorms from June to October. The annual rainfall is around 1660 mm.

144-> roughly three storm surges have occurred in
Line140, we rephrased the sentence as follows:

Storm tide flooding caused by TCs is one of the main natural hazards in Haikou, roughly three storm surges have occurred in Haikou every year in recent decades.

144-> For example, during Typhoon Kalmaegi (2014), a total of 219.8 mm (24h)
Line42-43, we rephrased the sentence as follows:

For example, during Typhoon Kalmaegi (2014), a total of 219.8 mm (24h) of precipitation were produced and the highest tide level reached 4.3 m in Haikou.

178-> Delft3D Flexible Mesh (DFM), developed by Deltares in 2011, is a practical unstructured
Line177, we rephrased the sentence as follows:

Delft3D Flexible Mesh (DFM), developed by Deltares in 2011, is a practical unstructured shallow water flow calculation model.

179-> It can be used for ocean hydrodynamic and surface
Line178, we rephrased the sentence as follows:

It can be used for ocean hydrodynamic and surface runoff numerical simulations.

181-182-> needed to estimate the overland flow boundary and simulate the overland inundation during the TCs period
Line179, we rephrased the sentence as follows:

In this study, the DFM model was established to calculate the hydraulic boundary conditions needed to estimate the overland flow boundary and simulate the overland inundation during the TCs period (Gori et al., 2020b).

Reference:
Gori, A., Lin, N., Xi, D.: Tropical cyclone compound flood hazard assessment: from investigating drivers to quantifying extreme water levels. Earths Future, 8, https://doi.org/10.1029/2020EF001660, 2020b.

187-189-> The minimum mesh grid size is 100 m, and the maximum mesh grid size is 12000 m. The astronomical tide is simulated by importing the phase
Line186, we rephrased the sentence as follows:

The minimum mesh grid size is 100 m, and the maximum mesh grid size is 12000 m. The astronomical tide is simulated by importing the phase and amplitude of tidal constituents (Q1, P1, O1, K1, N2, M2, S2, and K2) extracted from the global tidal model (TPXO8.0).

194-> The storm surge model is validated against the measured astronomical tide
Line192, we rephrased the sentence as follows:

The storm surge model is validated against measured astronomical tides and storm tides (astronomical tide plus storm surge).

208-212-> We collect the inundation data of TC1415 and conduct fieldwork in Haikou to validate this model. The overland inundation model can be approximately validated by comparing the inundation map of TC1415 with measured inundation area and depth.
Line208, we rephrased the sentence as follows:

We collect the inundation data of TC1415 and conduct fieldwork in Haikou to validate this model. The overland inundation model can be approximately validated by comparing the inundation map of TC1415 with measured inundation area and depth.

217-> Therefore, 66 TCs from 1960 to 2017 are selected
Line218, we rephrased the sentence as follows:

Therefore, 66 TCs from 1960 to 2017 are selected in this study (Figure 2), and we construct typhoon wind fields and simulate the storm tide of these TCs.

240-241-> which equals 1/P, to investigate the possibility of an extreme storm tide. The corresponding TC events in 5-, 10-, 25-, 50-, and 100-year return periods can be found to
Line240, we rephrased the sentence as follows:

We replace P with storm tide return periods (T), which equals to 1/P, to investigate the possibility of an extreme storm tide. The corresponding TC events in 5-, 10-, 25-, 50-, and 100-year return periods can be found to compare the compound flood hazards with different storm tides.

266-267-> We utilize the TC1415 event also to validate the overland flooding model.
Line267, we rephrased the sentence as follows:

We utilize the TC1415 event also to validate the overland flooding model.

270-271-> the actual flood distribution. Hence this overland flooding model has a good capacity for modelling and demonstrating TC flood hazards
Line270, we rephrased the sentence as follows:

Furthermore, the distribution of simulated inundation area is also consistent with the actual flood distribution. Hence this overland flooding model has a good capacity for modelling and demonstrating TC flood hazards.

318-319-> For a 100-year TC event, the inundation area with a depth above 1.0 m increases by approximately 2.5 times compared with a 5-year TC event
Line319, we rephrased the sentence as follows:

For a 100-year TC event, the inundation area with a depth above 1.0 m increases by approximately 2.5 times compared with a 5-year TC event.

334-> Figure 8 compares the overall inundation area of rainstorm
Line335, we rephrased the sentence as follows:

Figure 8 compares the overall inundation area of rainstorm, storm tide, and compound flooding under different return periods.

350-351-> This study investigates the compound effect of flood hazards
Line351, we rephrased the sentence as follows:

This study investigates the compound effect of flood hazards by studying the probability distribution of highest storm tides during TCs.

355-357->Copula function has been confirmed to model and describes the dependence between flood variables and express compound flood risk
Line358, we rephrased the sentence as follows:

In recent years, copula function has been confirmed to model and describe the dependence between flood variables and express compound flood risk (Zellou and Rahali, 2019).

Reference:
Zellou, B., Rahali, H. Assessment of the joint impact of extreme rainfall and storm surge on the risk of flooding in a coastal area. J. Hydrol., 569, 647–665, https://doi.org/10.1016/j.jhydrol.2018.12.028, 2019.

375-377Although this study is limited to Haikou City, the methodology of quantitatively assessing compound flooding risks through constructing a coupled framework of two hydrodynamic models is available for other coastal cities. ->Hard to read sentence.
Line384, we rephrased the sentence as follows:

Although this study is limited to Haikou City, we confirmed that it is available for other coastal cities to adopt the methodology of coupling two hydrodynamic models to quantitatively assessing compound flooding risks.

This paper presents a study on compound flooding in coastal regions. Using Haikou as a case study area, the authors couple a storm surge model and overland flooding model based on Delft3D Flexible Mesh model to investigate the compound effect of tropical cyclone flood hazards. It is an interesting and well-written paper. This paper is on a topic of interest to the audience of NHESS. The modeling and analysis methods are scientifically sound. The results provide helpful insights about coastal compound floods. I only have a few minor comments that I hope the authors could address in their revision:

Specific comments:

1. It would be helpful to have a figure showing location of tide and rainfall gauges in the bigger graph of figure 1. I am not very familiar with the geography of the region and I suspect many readers may not be either.
We have modified the figure to show the location of tide and rainfall gauges in the paper.

2. Section "TCs influencing Haikou" on lines 212-215: The TCs that pass through the region (18-22°N, 109-113°E) and stay over 24 hours have an apparent effect on Haikou. Therefore, 66 TCs from 1960 to 2017 are selected in this study (Figure 2), …. I suggest that the authors explain clearly about the selection criteria.
Thanks for the suggestions. We included more details on this in the revision:
Line 214: The TCs that pass through the region (18-22°N, 109-113°E) and stay over 24 hours have an apparent effect on Haikou (Ding, 1999; Wang, 1998; He, 1988). According to this, we analyze historical TC tracks and give the priority to the TC that passing between latitudes 18°N and 22°N and longitudes 109°E and 113°E. TC tracks lasting less than 24 hours in this region are removed in this study. Therefore, 66 TCs from 1960 to 2017 are selected in this study (Figure 2), and we construct typhoon wind fields and simulate the storm tide of these TCs. Each TC event has a code, for example, the ninth typhoon in 1963 is coded as TC6309.

3. It would be interesting to see the impact of climate change on compound flood. The authors may add some discussion related to this topic.
Thanks for the suggestions. We discussed more about the impact of climate change on compound flood.
Line 388: For future research on extreme TC compound flooding, climate change factors should be taken into consideration, such as sea level rise and land subsidence, and copula function can be applied to study the statistical dependence between heavy rainfall and strong storm surge under the changing environment to reveal extreme flood risk in coastal cities.

4. This paper conducts a probability distribution of storm tide, while doesn't consider the rainfall probability distribution. I think it is one of limitations in this study, the authors should give some additions about this limitation in the conclusion part.
Thanks for the suggestions. We have included this in the paper as follows:
Line 380: In this study, we selected the typical TC scenarios based on storm tide probability distribution. The high storm tide has been confirmed to be the main driving factor of flooding in previous studies (Xu et al., 2019, 2018; Lian et al., 2017). Considering that rainfall is also the driving factor of TC compound flooding, we will focus on the joint probability distribution of rainfall and storm tide in future research.

References:
Xu, H., Xu, K., Lian, J., et al.: Compound effects of rainfall and storm tides on coastal flooding risk. Stoch. Environ. Res. Risk Assess, 33, 1249–1261, https://doi.org/10.1007/s00477-019-01695-x, 2019.
Xu, H., Xu, K., Bin, L., et al.: Joint Risk of Rainfall and Storm Surges during Typhoons in a Coastal City of Haidian Island, China. Int. J. Environ. Res. Public. Health, 15, 1377, https://doi.org/10.3390/ijerph15071377, 2018.
Lian, J., Xu, H., Xu, K., et al.: Optimal management of the flooding risk caused by the joint occurrence of extreme rainfall and high tide level in a coastal city. Nat. Hazards, 89, 183–200,

https://doi.org/10.1007/s11069-017-2958-4, 2017.